# Investigating Vα7.2^+^/CD161^−^ T Cell and MAIT Cell Profiles Using Flow Cytometry in Healthy Subjects and Subjects with Atopic Dermatitis

**DOI:** 10.3390/ijms25063486

**Published:** 2024-03-20

**Authors:** Parvind Singh, Krisztian Gaspar, Andrea Szegedi, Laszlo Sajtos, Sandor Barath, Zsuzsanna Hevessy

**Affiliations:** 1Department of Laboratory Medicine, Faculty of Medicine, University of Debrecen, 4032 Debrecen, Hungary; parvind.singh@med.unideb.hu (P.S.); barath.sandor@med.unideb.hu (S.B.); 2Department of Dermatology and Venereology, Faculty of Medicine, University of Debrecen, 4032 Debrecen, Hungary; gaspar.krisztian@med.unideb.hu (K.G.); aszegedi@med.unideb.hu (A.S.); sajtos.laszlo@med.unideb.hu (L.S.)

**Keywords:** MAIT cells, Vα7.2^+^/CD161^−^ T cells, flow cytometry, dimensionality reduction, unsupervised clustering, atopic dermatitis

## Abstract

This study investigates the roles of mucosal-associated invariant T (MAIT) cells and Vα7.2^+^/CD161^−^ T cells in skin diseases, focusing on atopic dermatitis. MAIT cells, crucial for bridging innate and adaptive immunity, were analyzed alongside Vα7.2^+^/CD161^−^ T cells in peripheral blood samples from 14 atopic dermatitis patients and 10 healthy controls. Flow cytometry and machine learning algorithms were employed for a comprehensive analysis. The results indicate a significant decrease in MAIT cells and CD69 subsets in atopic dermatitis, coupled with elevated CD38 and polyfunctional MAIT cells producing TNFα and Granzyme B (TNFα^+^/GzB^+^). Vα7.2^+^/CD161^−^ T cells in atopic dermatitis exhibited a decrease in CD8 and IFNγ-producing subsets but an increase in CD38 activated and IL-22-producing subsets. These results highlight the distinctive features of MAIT cells and Vα7.2^+^/CD161^−^ T cells and their different roles in the pathogenesis of atopic dermatitis and provide insights into their potential roles in immune-mediated skin diseases.

## 1. Introduction

While conventional T cells recognize peptide antigens through the major histocompatibility complex (MHC), unconventional T cells are distinct, being non-MHC-restricted T cells that recognize nonpolymorphic, nonpeptide antigens. Based on antigen recognition, they are characterized as CD1a, CD1b, CD1c, CD1d, HLA E, and nonpolymorphic MHC class I related-1 molecule (MR1) cells. MR1-restricted cells are referred to as mucosa-associated invariant T (MAIT) cells [1].

MAIT cells are a highly conserved T cell subset found in mammalian species, and they are characterized by the expression of a semi-invariant αβ T cell receptor (TCR) composed of a Vα7.2 (TRAV1-2)/Jα33 α-chain paired with either Vβ13 (TRBV6) or Vβ2 (TRBV20) β-chains [2]. Presently, human MAIT cells are identified either with 5-OP-RU-loaded MR-1 tetramers or CD161^++^ together with TCR Vα7.2-Jα33^+^ [3]. Within the Vα7.2^+^ T cell compartment, the CD161^−^ subset can also be identified. Human C-type lectin-like CD161 is a type-II transmembrane protein and is expressed on about 25% of human adult T cells, including CD4 and CD8 [4]. Overall, CD161 expression is linked to memory cells. Within the CD4 compartment, CD161^+^ and CD161^−^ can be identified, whereas in the CD8 compartment, CD161 can be classified as CD161^−^, CD161^+^, and CD161^++^ [5,6].

Within the Vα7.2^+^ T cell compartment, the gene expression profiles of CD161^−^ (Vα7.2^+^/CD161^−^ T cells) and CD161^+^ (MAIT cells) cells are different. However, they share some common upregulated and downregulated genes that are different from those in conventional T cells and might provide a clue concerning the relationship between these cells [7]. MAIT cells recognize the microbial riboflavin-derivative antigens that are presented by the nonpolymorphic MHC class I-like protein MR1 [8] and are abundant in the liver, representing up to 45% of liver T cells [9] and up to 10% of T cells in peripheral blood (PB) [10]; no significant difference was found between sex (male and female) [11]. The roles of Vα7.2^+^/CD161^−^ T cells are not clearly understood. However, a significant increase in the PB was found in chronic HCV patients compared to healthy controls, whereas MAIT cells were compromised [12]. A similar finding was published in the case of patients with a chronic HIV-1 infection, where Vα7.2^+^/CD161^−^ accumulated and MAIT cells decreased compared to in uninfected healthy controls, despite an unchanged total T cell number [13]. The active roles of MAIT cell subsets were identified in the pathogenesis of skin diseases, such as psoriasis and hidradenitis suppurativa, where they were found to be another alternative source of IL-17, increasing the severity of disease [14,15].

Several subsets of T cells participate in atopic dermatitis (AD) pathogenesis. The inflammatory environment is a result of cytokine-mediated crosstalk between dendritic cells; Langerhans cells; mast cells; and immune cells, mainly T helper (Th) 2, Th22, Th17, and Th1 cells and innate lymphoid cells 2 (ILC2). The pathophysiology of the disease is complex and involves a multifactorial process, including skin barrier disruption and dysfunction, microbiota dysbiosis, IgE-mediated hypersensitivity, and a dysregulated immune system response [16]. Skin barrier disruption leads keratinocytes to recruit ILC2 and Th2 cells through chemokines. Allergen-activated dendritic cells induce naïve T cells to produce IL-4, and IL-13 and ILC2 produce IL-5 and IL-13. Together, these type 2 cytokines promote immunoglobulin E (IgE) class switching, eosinophil recruitment, and Th22 differentiation, leading to skin barrier disruption and a pro-inflammatory environment. In the acute phase, Th2/22 cells drive strong type 2 inflammation, while in the chronic phase, the predominant inflammation is supported by Th1 cells [17,18,19,20].

The exact contribution of MAIT cells and Vα7.2^+^/CD161^−^ T cells to human AD remains unknown. Eosinophils and basophils are two of the major contributors to type 2 inflammation in AD, and, in their absence or a functional MAIT cell blockade, skin-infiltrating eosinophils and IL-4-producing basophils were not able to be activated in a mouse model; as a result, a decrease in inflammation was noted [21]. Nevertheless, very little data are available on MAIT cells, no data are available for Vα7.2^+^/CD161^−^ T cells in human AD, and the role of these cells is unknown. We characterize MAIT and Vα7.2^+^/CD161^−^ T cells in PB by flow cytometry in newly diagnosed AD patients using conventional two-dimensional gating and a multiparametric multidimensional data analysis using machine learning algorithms. Also, we discuss the differences between MAIT and Vα7.2^+^/CD161^−^ T cells.

## 2. Results

### 2.1. Cellular Surface Marker Characterization of T Cells, MAIT Cells, and Vα7.2^+^/CD161^−^ T Cells in AD

Two-dimensional sequential gating was used to analyze the cellular characteristics of T cells and the Vα7.2^+^ compartment containing Vα7.2^+^/CD161^+^ (MAIT cells) and Vα7.2^+^/CD161^−^ T cells. T cells (% of lymphocytes) were found to be significantly higher in AD than in HCs, along with CD38^+^, which is an activated T cell subset (Figure 1A). In contrast to T cells, the MAIT cell percentage (% of T cells) decreased. However, CD38^+^, which is an activated MAIT cell subset, was higher (Figure 1B). Like MAIT cells, the Vα7.2^+^/CD161^−^ T cell percentage (% of T cells) decreased and the CD38^+^ subset increased in AD (Figure 1C). Other surface markers, namely, CD69, PD-1, CLA, CD4, and CD8, did not show any significant difference in T cells, MAIT cells, or Vα7.2^+^/CD161^−^ T cells (Appendix A).

### 2.2. Type of T Cell, MAIT Cell, and Vα7.2^+^/CD161^−^ T Cell Response in AD

The Th2/22 dominance of T cells is a classical feature of AD, and our results also show a significantly lower percentage of Interferon γ (IFNγ) production and slightly upregulated however nonsignificant IL-4 producing T cells. A higher percentage of IL-22-producing T cells and no significant difference in IL-17A-producing T cells were found (Figure 2A). A polyfunctional double-positive Tumor Necrosis Factor α (TNF α)/Granzyme B (GzB)-producing MAIT cell subset was significantly higher in AD (Figure 2B), suggesting the involvement of PB MAIT cells in disease pathogenesis. Conversely, single-positive GzB and TNFα, along with IFNγ, did not show any significant difference. The Vα7.2^+^/CD161^−^ T cell response was similar to the conventional T cells’ pattern. Th1 (IFNγ)-producing cells were decreased and slightly upregulated but with non-significant IL-4 production, and a significant upregulation of Th22 (IL-22)-producing cells was found. IL-17A-producing Vα7.2^+^/CD161^−^ T cells did not show any significant difference (Figure 2C). Other intracellular markers—like IL-17A, IL-22, and IL-18Rα of MAIT cells; TNFα, GzB, TNFα/GzB, and IL-18Rα of T cells; and Vα7.2^+^/CD161^−^ T cells—did not show any significant difference (Figure 2A–C).

### 2.3. Comparison of MAIT and Vα7.2^+^/CD161^−^ T Cells 

To find out the difference between MAIT and Vα7.2^+^/CD161^−^ T cells, we compared MAIT cells and their subsets with Vα7.2^+^/CD161^−^ T cells and their subsets in AD and HCs separately. In HCs, the comparison of all datasets showed significant differences, except for IL-17A and GzB/TNFα subsets of MAIT cells and Vα7.2^+^/CD161^−^ T cells. Within the Vα7.2^+^ compartment, Vα7.2^+^/CD161^+^ cells (MAIT cells) were higher than Vα7.2^+^/CD161^−^ T cells. MAIT cells were mainly CD69^+^ rather than CD38^+^, while Vα7.2^+^/CD161^−^ T cells were the opposite. PD-1 expression was also significantly higher in MAIT cells, and majority of them were CD8^+^ cytotoxic T cells. Conversely, Vα7.2^+^/CD161^−^ T cells showed more of a CD4^+^ helper phenotype and CLA expression. MAIT cells expressed higher IFNγ and lower IL-4, and non-significant differences were found in IL-17A compared to Vα7.2^+^/CD161^−^ T cells. IL-22 production of MAIT cells was higher while, GzB and TNFα/GzB production of MAIT cells was lower compared to Vα7.2^+^/CD161^−^ T cells. The classical conclusion that MAIT cells are different from Vα7.2^+^/CD161^−^ T cells is based on IL-18Rα expression; IL-18Rα and TNFα both showed a significantly higher percentage in MAIT cells. However, in the AD comparison, only the subsets affected by AD pathogenesis were different. MAIT cells (% T cells), which were downregulated in AD, became non-significant. IL-22-producing Vα7.2^+^/CD161^−^ T cells were higher in AD and became non-significant. Polyfunctional TNFα/GzB-producing MAIT cells were upregulated in AD and became significant (Figure 3).

### 2.4. Multidimensional T Cell Analysis 

A T cell multidimensional analysis of tubes 1, 2, and 3 revealed several immune clusters that were significantly different between AD patients and healthy controls. Based on the expression of the markers in the tubes, the phenotypes of the immune clusters were determined. Overall, T cells, CD161^+^/T cells, Vα7.2^+^/CD161^−^ T cells, MAIT cells, CD69^+^/MAIT cell, CD38^+^/T cells, CD4^+^/CD161^+^/T cells, CD8^+^/Vα7.2^+^/CD161^−^ T cells, CD38^+^/Vα7.2^+^/CD161^−^ T cells, CD4^+^/MAIT, CD8^+^/MAIT, DN/MAIT cells, IFNγ^+^/T cells, IL-22^+^/T cells, and IFNγ^+^/Vα7.2^+^/CD161^−^ T cells were found to be statistically significantly different (Figure 4). There was a higher representation of CD3^+^ T cells and a lower representation of CD161^+^ T cells, MAIT cells, CD69^+^ MAIT cell subsets, and Vα7.2^+^/CD161^−^ T cell subsets in AD than in HCs (Figure 5A). Tube 2 revealed further subsets of these cells, based on cellular surface expression. CD4^+^/CD161^+^ T cells were lower, CD38^+^ T cells were higher, MAIT cell subsets (CD4, CD8, and DN) were lower, CD8^+^/Vα7.2^+^/CD161^−^ T cells were also lower, and CD38^+^-activated Vα7.2^+^/CD161^−^ T cells were higher in AD (Figure 5B). Data from tube 3 revealed a Th2/22 response of T cells, where IL-22-producing CD3^+^ T cells were higher and IFNγ-producing cells were lower. IFNγ-producing Vα7.2^+^/CD161^−^ T cells were also found to be suppressed in AD (Figure 5C). We did not find any significant difference in tube 4 using the multidimensional analysis.

## 3. Discussion

Conventional T cells have been extensively studied in different diseases, whereas the roles of subsets such as MAIT cells are not fully understood, and little data are available for Vα7.2^+^/CD161^−^ T cells. Our study aimed to investigate the significance of MAIT and Vα7.2^+^/CD161^−^ T cells in peripheral blood samples of healthy control subjects and patients with early-stage AD. Therefore, we investigated various cell surface and intracellular markers on these cells.

The T cell percentage of lymphocytes was significantly higher, while lower MAIT and Vα7.2^+^/CD161^−^ T cell percentages were found in AD. MAIT cells recognize microbial vitamin B2 (riboflavin) metabolites from a wide range of microbes presented by MR1 molecules [22]. It has been known for a long time that Staphylococcus aureus infection exacerbates AD [23]; and the riboflavin biosynthesis pathway of *S. aureus* generates MR1 antigens, which are recognized by MAIT cells [24]. Staphylococcal enterotoxin causes a cytokine storm, which is also supported by MAIT cells. However, after a cytokine storm, MAIT cells become unresponsive, similar to an anergy state. It was hypothesized that *S. aureus* hijacks MAIT cells during a cytokine storm and leaves them functionally impaired [25]. Recently, MAIT cell recruitment and function were described in wound healing. MAIT cell recruitment to the wound site is CXCR-6- and CXCL-16-dependent and independent of MR-1 activation. Upon recruitment, MAIT cells produced tissue repair factors such as amphiregulin in a mouse model [26]. Cutaneous lymphocyte-associated antigen (CLA) is a unique skin-homing receptor that facilitates the targeting of T cells to inflamed skin. Skin-homing T cells recirculate between skin sites of inflammation and the bloodstream, allowing us to consider the cells in the bloodstream a potential surrogate for those in the skin. This hypothesis was supported by our finding that Vα7.2^+^/CD161^−^ T cells showed an upregulated expression of CLA in AD patients compared to healthy controls; however, the difference was not significant. In contrast, the significant depletion of Vα7.2^+^/CD161^−^ T cells and MAIT cells in peripheral blood might be explained by their infiltration and accumulation to sites of inflammation in the skin. We did not find CLA expression on MAIT cells in peripheral blood. It is possible that, instead of CLA-based recruitment and infiltration to inflamed skin, MAIT cells migrate with the help of different chemokine receptors, such as CXCR-6 and CXCL-16, as described above. This might explain the lower number of MAIT cells in our AD cases in PB since they might migrate to skin lesions. However, it is insufficient to conclude the role of MAIT cells since we did not study the skin lesions of these patients in parallel. Notably, CD38 is a better activation marker in the case of AD than CD69, as it was found to be upregulated in all subsets (T cells, MAIT, and Vα7.2^+^/CD161^−^ T cells). CD38 is an activation marker expressed in thymocytes, T and B lymphocytes, the myeloid lineage, natural killer T cells, and granulocytes [27]. CD38 has been reported as a chronic activation marker that can be used to monitor the activation status in extrinsic AD [28]. CD4^+^/CD38^+^ T cells were found to be part of a compromised polyfunctional response to Staphylococcal antigens in an ex vivo experiment in chronically activated AD patients [29].

The classical Th2/22 response could be observed in our AD patients. IL-22-producing T cells were upregulated, and Th1 (IFNγ)-producing T cells were downregulated. Th2/22 cell-driven type 2 inflammation is a classical feature of AD [17,18,19,20]. Significantly higher amounts of IL-22-producing T cells are distinct in AD patients [30]. Keratinocytes express IL-22 receptors widely, and studies have shown that IL-22-keratinocyte crosstalk plays an essential role in skin barrier defense and AD pathogenesis [31]. Real-time PCR data revealed that the most prominent gene activation of Th22 immune pathways was found in skin samples of the acute phase and then progressively increased during the chronic phase of the disease [32]. Polyfunctional GzB/TNFα-producing MAIT cells were upregulated and could be detected at a higher percentage in our AD patients. MAIT cells are polyfunctional, and GzB^+^TNFα^+^-producing MAIT cells were found to be significantly lower in the female genital tract upon E. coli stimulation compared to in blood [33]. GzB has been found to be an important factor involved in AD pathogenesis [34]. Emerging studies suggest that GzB is involved in epithelial barrier dysfunction via the proteolytic cleavage of E-cadherin in the epidermis and filaggrin within the stratum corneum [35]. Increased TNFα membrane receptors on the immune cells of AD patients have been found [36]. 

In our study, IFN-γ producing Vα7.2^+^/CD161^−^ T cells showed a lower representation and IL-22-producing Vα7.2^+^/CD161^−^ T cells showed a higher representation in AD. The suppression of Th1-type Vα7.2^+^/CD161^−^ T cells and the upregulation of Th22 cells suggest that these cells also contribute to Th2/22-mediated inflammation in AD. The role of these T cell populations has not been established yet. They might be undifferentiated MAIT cells or could be the result of CD161 downregulation. A significant increase in PB Vα7.2^+^/CD161^−^ T cells was noted in patients with chronic HCV and chronic HIV-1 infections compared to healthy controls, whereas MAIT cells were compromised [12,13].

To find out the similarity between Vα7.2^+^/CD161^+^ (MAIT cells) and Vα7.2^+^/CD161^−^ T cells in HCs and AD, we compared their immunophenotypes. In HCs, overall, Vα7.2^+^/CD161^−^ T cells expressed more IL-4 and GzB than MAIT cells. Interestingly, they expressed more CD38 and less CD69 activation markers than MAIT cells. In terms of cytotoxic and helper T cells, Vα7.2^+^/CD161^−^ T cells were represented as helper T cells rather than cytotoxic T cells, while MAIT cells were mainly CD8^+^. Vα7.2^+^/CD161^−^ T cells expressed very low levels of IL-18Rα compared to MAIT cells; this is the best evidence that they are different from MAIT cells, as IL-18Rα is a surrogate marker of MAIT cells, which are regulated by promyelocytic leukemia zinc finger (PLZF) and RAR-related orphan receptor gamma (ROR-gamma) transcription factors [9]. RNA sequencing of a similar phenotype by Park et al. showed that TCR Vα7.2^+^CD161^−^ T cells are markedly different from MAIT cells and highly similar to conventional T cells, despite the expression of Vα7.2. However, MAIT and Vα7.2^+^/CD161^−^ cells share some common upregulated and downregulated genes [7]. Kathrin et al. showed CDR3’s repertoire of Vα7.2^+^ chains and found a dominating clonal expansion of canonical and noncanonical clones within MAIT cells, whereas Vα7.2^+^/CD161^−^ subsets were polyclonal [37]. In AD patients, a comparison of MAIT and Vα7.2^+^/CD161^−^ T cell subsets gave the same results as in HCs, except for in terms of the MAIT and Vα7.2^+^/CD161^−^ T cells themselves and their subsets IL-22 and GzB/TNFα. The reason for these differences is due to the decrease in total MAIT cells, increase in GzB/TNFα-producing MAIT cells, and increase in IL-22-producing Vα7.2^+^/CD161^−^ T cells in AD, as described above.

A multidimensional analysis revealed similar and comparable results overall. T cells and CD38^+^ subsets of T cells were significantly higher in AD, with higher Th22 and lower IFN-γ. Overall, MAIT cells and their subsets CD4^+^, CD8^+^, DN, and CD69^+^ were affected and decreased in AD. Vα7.2^+^/CD161^−^ T cells and their CD8^+^ subsets, along with IFN-γ-producing Vα7.2^+^/CD161^−^ T cells, were decreased in AD, and the chronically activated CD38^+^ subset was significantly higher in AD. Major components of T cells were commonly identified using both methods. The reason for the discrepancy between the results obtained from the traditional and multidimensional analyses might be due to the multidimensional nature of data. All the fluorescence signals are considered during the clustering process, unlike during traditional two-dimensional gating, where only one fluorescence signal is considered at a time, and, based on that, the gates are decided. 

A multidimensional analysis was performed on downsampled T cells, and 20,000 randomly selected T cells were used from each sample. Immune clusters of different phenotypes are presented as the percentage of T cells, as they came from the parent gate. The fractions of MAIT cells and Vα7.2^+^/CD161^−^ T cells were already very small compared to the total T cells. Further, the phenotypes of MAIT cell and Vα7.2^+^/CD161^−^ T cell fractions were even less represented. It is possible that smaller populations of MAIT cells and Vα7.2^+^/CD161^−^ T cells were missed, owing to their small representation from the total T cell gate, unlike in two-dimensional gating, where they are represented from parent gates. However, it has been verified that machine learning algorithms avoid manual biased gating and potentially detect novel cell types and cellular relationships. These populations might be missed in traditional gating due to the complexity of cellular heterogeneity and the limitation of exploring all the dimensions of datasets at the same time [38,39,40,41]. In our case, the additional phenotypes that we identified using the multidimensional analysis were CD8^+^ subsets of Vα7.2^+^/CD161^−^ T cells; CD69^+^, CD4^+^, CD8^+^, and DN MAIT cells; and CD161^+^ and CD4^+^/CD161^+^ T cells. CD161^+^ T cells and their subsets expressing CD4 were evident. CD161 cells are highly associated with the memory phenotype (central or effector), and they are highly functional [42]. Our results show a lower representation of helper T cells in AD than in HCs, which is in line with previous publications [43,44]. 

There are very limited studies mentioning MAIT cells in human AD, and, to the best of our knowledge, there is no study mentioning Vα7.2^+^/CD161^−^ T cells in AD. Naidoo et al. investigated the role of MAIT cells in AD using a mouse model. They found that MAIT cells promote AD by promoting eosinophil activation and the recruitment of IL-4-producing cells. The genetic deletion of MR1 resulted in the blockade of disease progression [21]. However, another study by Cassius et al. focused on the peripheral blood of human AD patients. They showed no significant difference in the frequency of MAIT cells between AD patients and HCs [45]. The reason for this discrepancy might be the lower sample size. A limitation of our study is that we used peripheral blood samples only. An analysis with skin biopsies was not performed in parallel to clarify the presence of these phenotypes in the skin. Also, all these samples belonged to early diagnosed patients, and it has been shown that, in the non-lesional and acute phases of the disease, the quantity of immune cells is significantly lower than in the chronic phase [46]. 

Overall, the analysis of cell surface and intracellular markers showed that the two populations of Vα7.2^+^ cells, MAIT cells and CD161^−^ T cells, are significantly different. Based on the expression pattern of the investigated markers, CD161^−^ T cells seem to show more similarity to conventional T cells. When examining samples from AD patients, we found differences in both cell types compared to healthy subjects, suggesting that they may play a role in the pathomechanism of the disease.

## 4. Materials and Methods

### 4.1. Study Ethics 

This study was approved by the Institutional Review Board of the Faculty of Medicine, University of Debrecen, Hungary (DE RKEB/IKEB 5404–2020).

### 4.2. Study Participants 

PB samples from 14 newly diagnosed, untreated Caucasian AD patients and 10 healthy controls (HCs) were collected between August 2022 and June 2023. Clinically, the eczema area severity index (EASI) was calculated before sample collection, and eosinophil count was performed after sample collection to examine the activation of immune cells. Appendix A presents Student’s *t* test performed on eosinophils (%) between AD and HCs (*p* values < 0.0001), and the mean EASI score was found to be 30.05 ± 13.06.

### 4.3. Sample Collection and Complete Blood Counts 

For sample collection, 3 mL PB was collected in BD vacutainer tubes containing tri-potassium ethylene-diamine tetraacetic acid (K3-EDTA), and 6 mL of PB was collected in a BD vacutainer tube containing sodium heparin anticoagulant (Becton Dickinson, San Jose, CA, USA). Complete blood counts were performed using a Siemens ADVIA 2120i hematology analyzer (Siemens Healthcare GmbH, Erlangen, Germany).

### 4.4. Peripheral Blood Mononuclear Cell (PBMC) Isolation and In Vitro Stimulation

PBMCs were isolated from heparin tubes by density gradient centrifugation. The PBMC count was adjusted to approximately 2 × 10^6^ cells/100 μL. PBMCs (300 μL of diluted PBMC) were stimulated with 30 ng/mL of phorbol 12-myristate 13-acetate (PMA) and 1 μg/mL of Ionomycin. The total volume was adjusted to 1 mL by adding RPMI medium and incubated for four hours at 37 °C in a 5% CO_2_ incubator in a FACS tube. Then, 20 μg/mL Brefeldin A was added for the last 3 h and 20 min of incubation.

### 4.5. Flow Cytometry 

A multiparametric eight-color flow cytometric experiment was performed using pre-titrated mouse anti-human monoclonal antibodies from Beckman Coulter (Brea, CA, USA), Biolegend (San Diego, CA, USA), BD/Pharmingen (Franklin Lakes, NJ, USA), and Exbio (Prague, Czech Republic): anti-human IL-1β (JK1B-1; Biolegend), anti-human cutaneous lymphocyte antigen (HECA-452; Biolegend); anti-human IFN-γ (25723.11; Biolegend) conjugated with fluorescein-isothiocyanate (FITC); anti-human IL-4 (3010.211; Biolegend), anti-human TNF-α (REA656; MACS), anti-human CD38 (HB-7; BD) conjugated with phycoerythrin (PE); anti-human TCR Vα7.2 (3C10; Biolegend) conjugated with peridinin chlorophyll protein/Cynine 5.5 (PerCP/Cy5.5); anti-human CD161 (191B8; BC/IOT) conjugated with PE-Cynine 7 (PC7); anti-human CD3 (SK7; BD), anti-human IL-18Rα (H44; Biolegend), anti-human IL-22 (2G12A41; Biolegend), anti-human CD279 (PD1) (REAL383; MACS) conjugated with allophycocyanin (APC); anti-human CD8 (SK1; BD), anti-human CD3 (SK7; BD) conjugated with allophycocyanin cyanine 7 (APC-H7); anti-human CD4 (RPA-T4; BD), anti-human IL-17A (BL168; Biolegend), anti-human granzyme B (QA16A02, Biolegend), anti-human CD69 (FN50; Biolegend) conjugated with pacific blue (PB); and anti-human CD45 (HI30; Exbio) conjugated with pacific orange (PO). These monoclonal antibodies were combined in four different tubes to characterize MAIT cells. Sample acquisition was performed using a BD FACSCanto II™ flow cytometer (Franklin Lakes, NJ, USA) and FACSDiva v8.0.2. software. A total of 300,000 events were recorded for each tube. For quality control, the cytometer setup and tracking beads (CS&T) were measured daily to maintain the performance tracking of the equipment. An external quality control assessment was also performed by participating in the UK-NEQAS immunophenotyping program.

#### 4.5.1. Surface Staining

PB collected in EDTA tubes was used for staining, lysis, and washing protocol-based surface immunophenotyping. Briefly, 100 µL PB was mixed with an appropriate cocktail of pre-titrated mAbs in a FACS tube and incubated for 15 min in the dark at room temperature (20–22 °C). Upon incubation, the cells were subjected to RBC lysis using lysis buffer (BD FACS^TM^ lysing solution) and incubated for 10 min. The tubes were then washed once with 1 mL of phosphate-buffered saline (PBS), centrifuged at 1500 rpm for 5 min, and resuspended in 400 µL of 1% paraformaldehyde.

#### 4.5.2. Intracellular Staining 

Intracellular staining was performed using an IntraPrep permeabilization reagent (Beckman Coulter Brea, CA, USA). Briefly, PBMCs were mixed with the appropriate concentrations of cellular surface antibodies and incubated for 15 min in the dark at room temperature. Subsequently, Intraprep permeabilization buffer 1 was added and incubated again for 15 min. The cells were then washed with 4 mL of PBS by centrifugation (1500 rpm for 5 min). Intraprep permeabilization buffer 2 was then added and incubated for 5 min (without vortexing). Subsequently, intracellular antibodies were added at appropriate concentrations and incubated for 30 min in the dark at room temperature. After incubation, the cells were washed again and resuspended in 400 μL of paraformaldehyde for acquisition.

### 4.6. Data Analysis

For each of the samples, multiparametric data were acquired from four different tubes. Tube 1 and tube 2 contained surface markers, whereas tubes 3 and 4 contained intracellular cytokines after PMA stimulation. Flowjo v10.9.0 (TreeStar Inc., Ashland, OR, USA) software was used for the sequential gating of T cells and their further subsets and a multidimensional data analysis.

#### 4.6.1. Sequential Gating of T Cells, MAIT Cells, and Vα7.2^+^/CD161^−^ T Cells

Universally, PeacoQC was applied to clean data, and then doublets were removed using FSC area vs. height and area vs. width. WBCs were gated as CD45^+^ events, and lymphocytes were derived from the WBC gate. Data were cleaned again to remove antibody precipitates on the CD45 vs. CD3 plot, and then T cells were gated within a clean gate. Within the T cell gate, Vα7.2^+^/CD161^++^ events were gated as MAIT cells, and Vα7.2^+^/CD161^−^ cells were gated as Vα7.2^+^/CD161^−^ T cells (Appendix A). Subsets CD4, CD8, PD-1, CD69, CD38, CLA, IFNγ, IL-4, IL-22, IL-17A, TNFα, GzB, TNFα/GzB, IL-1β, and IL-18Rα were gated within T cells, MAIT cells, and Vα7.2^+^/CD161^−^ T cells, and their percentage from the parent gate was used for statistical analyses.

#### 4.6.2. Multidimensional Data Analysis

All four different tubes, marked as tubes 1, 2, 3, and 4, are detailed in Table 1. The files (including AD and HCs) of tube 1 were analyzed together; similarly, all the files of tubes 2, 3, and 4 were analyzed together in separate multidimensional analyses. In total, we analyzed four multidimensional datasets for each of the tubes containing the same markers. The analysis began with gating out T cells, with the universal gate that is commonly used for this process (Appendix A). Next, 20,000 T cells were downsampled from each file (including AD and HC samples), and then downsampled gates based on CD3 expression were concatenated (merged) in a single file. This concatenated file contained the data of 200,000 T cells from each of the samples from the AD and HC groups. FlowJo plugins were used to run machine leaning algorithms on the concatenated file. Dimensionality reduction via Uniform Manifold Approximation and Projection (UMAP) was performed to visualize high-dimensional data [47]. The Phenograph clustering algorithm was run using the FlowJo plugin to refine and identify the number of clusters. Based on the clusters identified by Phenograph, FlowSOM was run to build self-organizing maps and identify meta-clusters of T cells [48]. FlowSOM meta-cluster numbers were optimized by clustering several times with different numbers of meta-clusters to avoid over- and underclustering (Appendix A). In tubes 1, 2, 3, and 4, ten, eighteen, fourteen, and sixteen immune clusters could be identified, respectively (Appendix A). We preferred slight overclustering to avoid missing less expressive phenotypes. The immune clusters identified by FlowSOM on the concatenated file (which contained the AD and HC samples) were then laid over individual samples of AD and HCs to compare the immune clusters between the two groups. Immune clusters presented as the percentage of T cells from individual samples (AD and HCs) were exported and compared with Student’s *t* test, and *p* values were calculated. The phenotype of the immune cluster showing significantly different *p*-values between AD and HCs was identified using a cluster explorer.

### 4.7. Statistical Analysis

All statistical analyses were performed using GraphPad Prism v9.0 (GraphPad Software, Inc., San Diego, CA, USA). Kolmogorov–Smirnov (SK) and Shapiro–Wilk normality tests were used to test the normality of data distribution. For non-normal distribution, the Mann–Whitney test was used, and, for normal distribution, Student’s *t* test was used; the statistical significance of the findings was set at a *p*-value of less than 0.05.

## Figures and Tables

**Figure 1 ijms-25-03486-f001:**
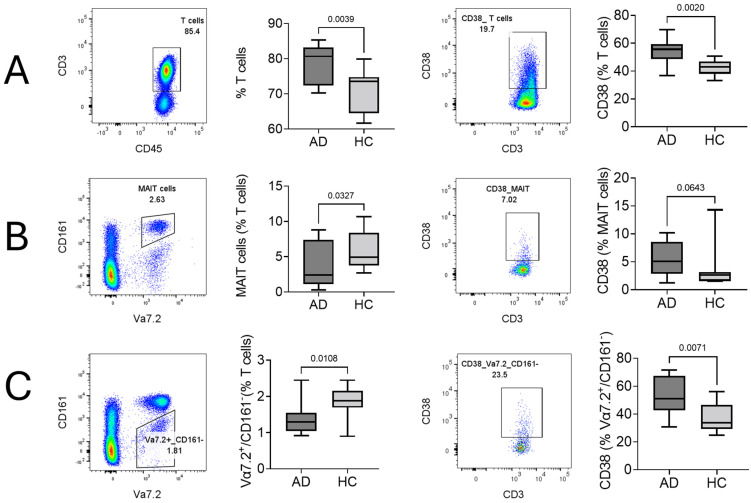
Significantly different expressions of cellular surface markers of T cells, MAIT cells, and Vα7.2^+^/CD161^−^ T cells between atopic dermatitis (AD) and healthy control (HC) subjects. Representative flow cytometric density dot plots along with their statistics are given. (**A**) T cells, (**B**) MAIT cells, and (**C**) Vα7.2^+^/CD161^−^ T cells. Student’s *t* test was used to calculate significant difference between AD and HCs. X-axis represents sample category (AD and HCs), and Y-axis represents % from T cells, MAIT cells, and Vα7.2^+^/CD161^−^ cells. *p* values are given within the plots.

**Figure 2 ijms-25-03486-f002:**
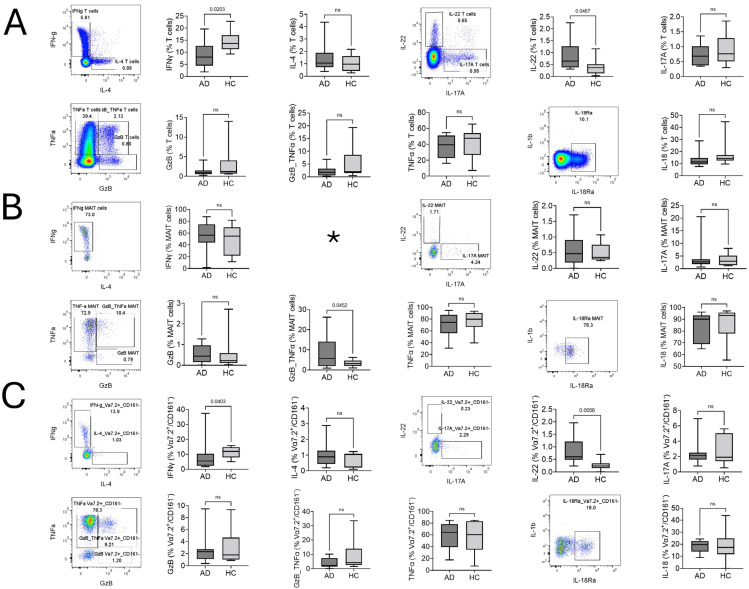
After phorbol 12-myristate 13-acetate stimulation, significantly different expression of intracellular cytokine-producing cells between atopic dermatitis (AD) and healthy control (HC) subjects. Representative flow cytometric dot plots along with their statistics are given. (**A**) T cells, (**B**) MAIT cells, and (**C**) Vα7.2^+^/CD161^−^ T cells. Student’s *t* test was used to calculate significant difference between AD and HCs. X-axis represents sample category (AD and HCs), and Y-axis represents % from T cells, MAIT cells, and Vα7.2^+^/CD161^−^ cells. *p* values are given within the plots; *p* values > 0.05 are represented as ns (non-significant). * IL-4-producing MAIT cell event number was below the limit of detection; hence, statistical calculation was not carried out.

**Figure 3 ijms-25-03486-f003:**
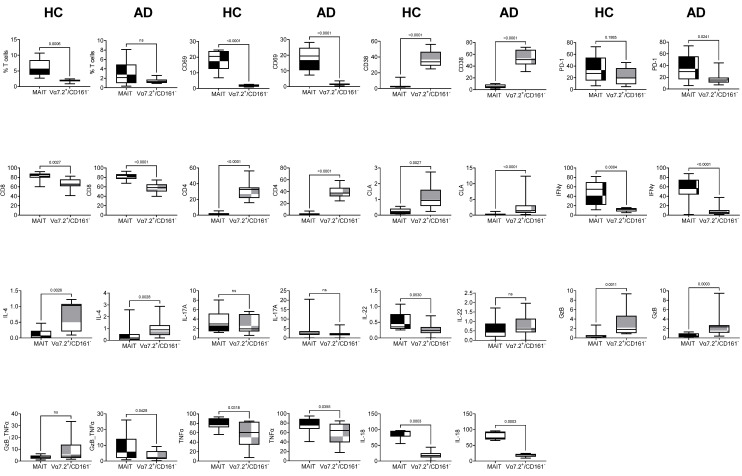
MAIT cells and Vα7.2^+^/CD161^−^ T cells and their subsets were compared in healthy control (HC) and atopic dermatitis (AD) samples. Student’s *t* test was used to calculate significant difference between MAIT and Vα7.2^+^/CD161^−^ T cells. X-axis represents MAIT cells and Vα7.2^+^/CD161^−^ T cells. Y-axis represents different subsets among MAIT and Vα7.2^+^/CD161^−^ T cells. *p* values are given within the plots; *p* values > 0.05 are represented as ns (non-significant). From top to bottom, HC and AD column comparisons are given in parallel.

**Figure 4 ijms-25-03486-f004:**
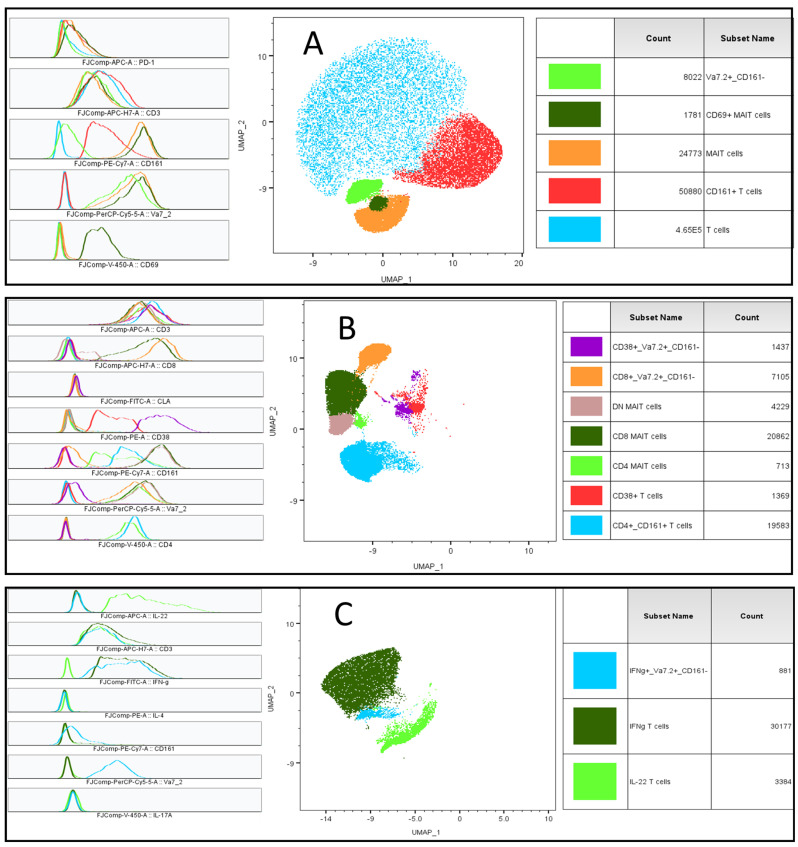
Dimensionality reduction (UMAP) and clustering (FlowSOM) of flow cytometric data are presented. T cell subsets that were significantly different between AD and HCs within tubes 1, 2, and 3 are presented in (**A**), (**B**), and (**C**), respectively. Middle plots show UMAP_1 and UMAP_2 plots; their phenotypes are shown in multiple overlay plots on the left side. The count of the total T cells and the names of subsets are given in the right-side panel.

**Figure 5 ijms-25-03486-f005:**
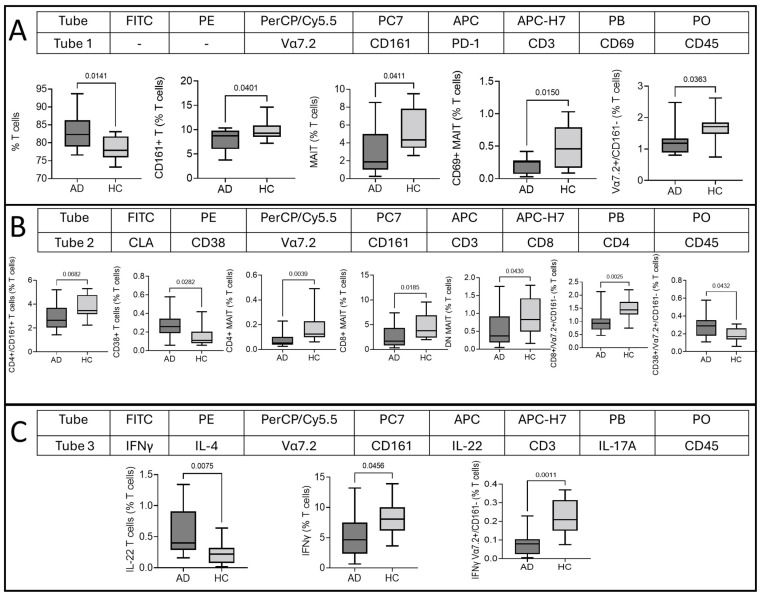
Significantly different subsets of T cells within healthy control (HC) and atopic dermatitis (AD) patients analyzed with multidimensional data analysis. Significantly different subsets within tubes 1, 2, and 3 are shown in (**A**), (**B**), and (**C**), respectively. We did not find any statistically significant results in tube 4 using multidimensional analysis. Monoclonal antibodies used in each tube are indicated. Subsets of T cells that were significantly different between AD and HCs were compared with Student’s *t* test and are presented in box plots. X-axis represents cells of AD and HCs. Y-axis represents different subsets among T cells. *p* values are given within the plots.

**Table 1 ijms-25-03486-t001:** Details of different tubes containing panels of monoclonal antibodies used for multicolor flowcytometric staining of samples, along with their fluorescent channels.

Tube	FITC	PE	PerCP/Cy5.5	PC7	APC	APC-H7	PB	PO
Tube 1	-	-	Vα7.2	CD161	PD-1	CD3	CD69	CD45
Tube 2	CLA	CD38	Vα7.2	CD161	CD3	CD8	CD4	CD45
Tube 3	IFN-γ	IL-4	Vα7.2	CD161	IL-22	CD3	IL-17A	CD45
Tube 4	IL-1β	TNF-α	Vα7.2	CD161	IL-18Rα	CD3	GzB	CD45

## Data Availability

The data that support the findings of the study are available from the corresponding author upon request.

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
