# Peer review of "Investigating Vα7.2+/CD161 T Cell and MAIT Cell Profiles Using Flow Cytometry in Healthy Subjects and Subjects with Atopic Dermatitis"

_ijms, 2024, doi:10.3390/ijms25063486_

Round 1

Reviewer 1 Report

Comments and Suggestions for Authors

The present paper explores the involvement of mucosal-associated invariant T (MAIT) cells and Vα7.2+/CD161- T cells in skin diseases, with a specific focus on Atopic Dermatitis (AD). MAIT cells were analyzed by flow cytometry alongside Vα7.2+/CD161- T cells in peripheral blood samples from 14 AD patients and 10 healthy controls. The results show a significant decrease in MAIT cells and CD69 subsets in AD, coupled with elevated CD38 and polyfunctional TNFα+/GzB+ MAIT cells. Vα7.2+/CD161- T cells in AD exhibited a decrease in CD8 and IFNγ-producing subsets, but an increase in CD38 activated and IL-22 producing subsets.

The topic of the manuscript is interesting as the exact contribution of MAIT cells and Vα7.2+/CD161- T cells in human AD remains unexplored. The study is well-articulated, presenting the data in a coherent manner. Despite these merits, a few concerns need to be addressed before the manuscript can be considered for publication in the International Journal of Medical Sciences (IJMS).

 Issue to address:

-In Figure 2, it is recommended to incorporate analogous analyses for three distinct cell populations: (A) T cells, (B) MAIT cells, and (C) Vα7.2+/CD161- T cells. The authors are encouraged to provide comprehensive results for each cell type, maintaining consistency across the presentation. Even in instances where statistical significance is not evident, it is essential to show comparable findings for all the investigated cell populations. This approach ensures a thorough and uniform representation of the data, contributing to the overall robustness of the analysis.

-Figures 3 and 4 could be condensed into a single, more concise figure, preferably used as Figure 1. This integration would improve overall clarity and streamline the visual representation of the data. Furthermore, it would contribute to a seamless flow in illustrating key results, facilitating a clearer and more organized presentation of the study findings.

-Figure 5-6-7. The clarity of the Multidimensional Data Analysis is lacking. It is not apparent what tubes 1, 2, 3, and 4 represent. A more comprehensive explanation is needed to elucidate how these tubes discriminate between cells from Atopic Dermatitis (AD) and Healthy Controls (HC). The Materials and Methods section requires further elaboration to provide a clearer understanding of the experimental setup. Furthermore, a figure summarizing the key findings from the existing figures 5, 6, and 7 is advisable. This summary figure should succinctly capture the main points, enabling a quick and comprehensive overview of the crucial results presented in the individual figures.

Comments on the Quality of English Language

Minor editing required

Reviewer 2 Report

Comments and Suggestions for Authors

I was pleased to read the results of this interesting study evaluating unconventional T cell subsets in the peripheral blood of 14 subjects with atopic dermatitis and 10 healthy controls. The methodology is sound, and the discussion/conclusion are coherent with the authors’ results.

To improve their study, I suggest the authors to consider the following issues:

- Please be consistent with abbreviations throughout the abstract and throughout the manuscript. For example, in the abstract you can leave “atopic dermatitis” without abbreviating to AD. In the text I suggest writing in full certain abbreviations on the first mention, such  as granzymeB (GzB)

- Line 47 “gender”: change to “sex”. (male, female = sex) - Please see https://www.who.int/europe/health-topics/gender#tab=tab_1

- Line 63 “they”: please clarify as it is unclear if you are referring to keratinocytes or Th2/ILC2

- Line 89 “CLA”: as you correctly point out in the discussion, a limitation of the present study is that pathogenic information on atopic dermatitis is derived from circulating lymphocytes and not from lesional skin of atopic dermatitis. However, skin homing T cells recirculate between skin sites of inflammation and the bloodstream, allowing us to consider cells in the bloodstream a potential surrogate for those in the skin. Could CLA+ subsets from your analysis help us support this concept? Conversely, there could be an imbalance with certain cell subsets that accumulate in skin sites of disease and are therefore depleted in the blood. 

- Figure 2. “PMA” please write the abbreviation in full in the figure legend.

- Author contributions: please check that you are using the terminology of the CREDIT taxonomy

- Informed consent statement: please state how informed consent was provided by the participating subjects.

Comments on the Quality of English Language

- Discussion: carefully check the syntax of several sentences and consider rewriting those that are too long to improve readability and clarity
